# When Test-Time Training Fails: A Critical Analysis of Robustness and Hyperparameter Sensitivity

**Ziqi Wang**
*Department of Computer Science*
*University of Illinois Urbana-Champaign*

*ziqiw9@illinois.edu*

**Xiusi Chen**
*Department of Computer Science*
*University of Illinois Urbana-Champaign*

**Gaotang Li**
*Department of Computer Science*
*University of Illinois Urbana-Champaign*

**Heng Ji**
*Department of Computer Science*
*University of Illinois Urbana-Champaign*

**Tong Zhang**
*Department of Computer Science*
*University of Illinois Urbana-Champaign*

**Reviewed on OpenReview:** *https://openreview.net/forum?id=OEh31N1Hoj*

## Abstract

Test-time training (TTT) via input perplexity minimization has emerged as a promising way to improve language model performance at inference time. However, its practical robustness and its applicability beyond popular benchmarks remain unclear. This paper presents a preliminary analysis of two questions: whether TTT is effective on unseen tasks, and how sensitive it is to hyperparameter choices. We evaluate TTT on three anti-memorization datasets—Memo-Trap, GSM-Symbolic, and Math-Perturb—using six models from the Qwen 2.5 and Llama 3 families. Our findings reveal that while TTT shows effectiveness on common benchmarks such as AIME 2024, it struggles with tasks designed to counter memorization. We are careful to separate two distinct claims: that TTT *fails* on anti-memorization tasks (which our experiments demonstrate directly) and *why* it fails. Data contamination of standard benchmarks is one plausible explanation, but our experiments do not isolate it from other factors such as task difficulty or distribution shift; we therefore treat contamination as a hypothesis rather than an established cause, and our strongest contamination-controlled signal comes from comparing AIME 2024 with the post-release AIME 2025. We identify substantial differences between optimizers, with SGD outperforming Adam despite its slower convergence. Through extensive hyperparameter sweeps over learning rate, number of training steps, weight decay, momentum, and gradient normalization, we show that TTT is highly sensitive to these choices, with no universal recipe across tasks and models. Notably, gradient normalization improves robustness by mitigating catastrophic performance drops and reducing sensitivity to the learning rate. We further find that tuning the feed-forward networks can achieve higher peak performance than full-model tuning, while tuning only the attention modules yields more stable worst-case performance. These findings underscore the need for further research into making test-time training practical and reliable for real-world deployment. Because this study focuses on a single TTT algorithm, input perplexity minimization, our conclusions may not extend to all TTT methods. We encourage the

community to pay closer attention to TTT's sensitivity so that it can be made better suited to real-world applications.

## 1 Introduction

Advocated by OpenAI o1 (Jaech et al., 2024), test-time scaling has attracted considerable attention in the language model (LM) community for its ability to enhance model capabilities across tasks ranging from everyday question answering to specialized domain reasoning. Test-time scaling spans a broad spectrum of techniques. One line of work scales the number of generated tokens, for example by eliciting the model's self-reflection ability (Muennighoff et al., 2025). Another incorporates training directly into the inference phase (Zuo et al., 2025; Hu et al., 2025b;a). We refer to the latter approach as test-time training (TTT).

TTT comprises a substantial body of research with diverse methodologies, spanning a range of algorithmic frameworks and modalities across multiple domains (Sun et al., 2020; Wang et al., 2025; Hardt & Sun, 2024; Sun et al., 2024; Dalal et al., 2025). These methods nevertheless share a common workflow:

Input $x \rightarrow$ Training model $\theta$ with TTT $f(x) \rightarrow$ Inference on target tasks $g(x)$

Here $g(x)$ denotes the target task to be evaluated, which may be a classification or a generation problem; in the era of LMs, $g(x)$ is unified as next-token prediction. The function $f(x)$ is the manually designed TTT objective and must be aligned with $g(x)$ (Sun et al., 2020). The design of $f(x)$ therefore varies across settings. For instance, $f(x)$ may be an image reconstruction task when $g(x)$ is image classification (Gandelsman et al., 2022), while Zuo et al. (2025) construct reward signals via majority voting and apply reinforcement learning on top of them.

Because LMs unify tasks into next-token prediction over a token sequence, a natural way to enable test-time training is to minimize the model's perplexity on the test input in a self-supervised manner, thereby adapting the model to the test distribution. This approach has proven effective both theoretically (Hu et al., 2025a) and empirically (Hu et al., 2025b). Despite these promising results, two concerns cloud its practical application.

First, existing evaluations focus predominantly on popular benchmarks. For example, Sun et al. (2020) evaluate on AIME 2024 (Mathematical Association, 2024), GSM8k (Cobbe et al., 2021), Math500 (Lightman et al., 2023), and GPQA (Rein et al., 2024), while Hu et al. (2025a) evaluate on GSM8k (Cobbe et al., 2021), MetaMATH (Yu et al., 2023), and Logiqa (Liu et al., 2020). Given the well-documented data contamination of current LMs (Wu et al., 2025), it remains unclear whether the reported gains stem from genuine distribution adaptation or from data contamination.

Second, unlike the conventional training workflow, TTT typically lacks a reliable validation set for selecting hyperparameters. Existing works generally adopt a single fixed configuration, which obscures how TTT would behave in real deployment. For example, Hu et al. (2025a) use the Adam optimizer (Kingma & Ba, 2014) with dataset-specific learning rates, and Hu et al. (2025b) use the Adam optimizer with a fixed learning rate, a fixed number of training steps, and weight decay. Since a validation set is rarely available for selecting hyperparameters at test time, it is important to understand and, where possible, reduce TTT's sensitivity to these choices.

Accordingly, this paper investigates two questions for input-perplexity-minimization TTT:

- Is TTT effective on unseen tasks?

- Is TTT sensitive to hyperparameters, and if so, how can this sensitivity be reduced?

To address the first question, we select three representative tasks specifically designed to resist the model's memorization: Memo-Trap (McKenzie et al., 2024), which asks the model to complete a famous quote with an uncommon word; and GSM-Symbolic (Mirzadeh et al., 2024) and Math-Perturb (Huang et al., 2025), which perturb problems from GSM8k (Cobbe et al., 2021) and MATH (Hendrycks et al., 2021b), respectively. To address the second question, we examine a range of hyperparameters: learning rate, number of training

steps, optimizer, weight decay, momentum, and gradient normalization. Experiments on six models from two model families (Qwen 2.5 (Qwen et al., 2025) and Llama 3 (Dubey et al., 2024)) yield the following findings:

- **Effectiveness and hyperparameter sensitivity**: Although TTT is effective on several popular benchmarks, we find that it struggles on datasets designed to resist memorization. We stress that this is an empirical finding about *failure*, which we distinguish from its *cause*: one possible explanation is that benchmark gains partly reflect the recall of memorized patterns rather than genuine adaptation, but our current experiments do not by themselves rule out alternatives (e.g., the anti-memorization tasks simply being more difficult). We therefore present contamination as a hypothesis to be tested rather than a demonstrated mechanism. Even on tasks where TTT is effective, the optimal learning rate and number of training steps vary across tasks.

- **Optimizers**: Although Adam typically converges faster, SGD is better suited to the loss landscape of this optimization problem. Specifically, SGD converges and overfits more slowly and attains more stable and higher performance than Adam. Even when Adam's learning rate is lowered to slow its convergence, its performance still trails SGD. Our preliminary analysis attributes this gap to the more accurate gradient direction afforded by SGD. Given the large performance gap, we adopt SGD when analyzing the remaining factors.

- **Regularization**: Regularization is commonly applied through weight decay; we experiment with a standard value of 0.01 and a large value of 0.5. Neither setting meaningfully reduces overfitting.

- **Gradient normalization**: Gradient normalization makes the model more robust to the choice of learning rate and number of training steps, substantially slows overfitting, and in some cases even improves TTT performance.

- **Momentum**: Momentum plays a role similar to gradient normalization but is less reliable, as it occasionally worsens TTT performance.

- **Parameter subsets**: We additionally tune only part of the model, namely the feed-forward networks (FFNs) or the attention modules. Attention modules have fewer parameters and are more robust to performance drops, whereas tuning FFNs can match or even exceed the peak performance of full fine-tuning.

In summary, this paper presents a preliminary analysis of the instability of test-time training for language models and argues for greater attention to making TTT practical. We emphasize that our study focuses on a single TTT objective, input perplexity minimization, so our conclusions may not extend to all TTT algorithms. We encourage the community to examine TTT's sensitivity more closely in order to make it better suited to real-world applications.

## 2 Method and Settings

### 2.1 Test-time Training

We denote the training dataset by $\mathcal{D}_{\text{train}} = \{(x_i, y_i)\}_{i=0}^{N}$ and the test dataset by $\mathcal{D}_{\text{test}} = \{(x_i)\}_{i=0}^{M}$. A language model $P_\theta(\cdot|x)$, parameterized by $\theta$, is trained by minimizing $\mathcal{L} = \sum_{(x,y)\sim\mathcal{D}_{\text{train}}} -\log(P_\theta(y|x))$, which corresponds to the standard pre-training and supervised fine-tuning objective.

A test-time training approach first defines a new objective that depends only on the model and the input, $\mathcal{L}_{\text{test}}(\theta, x)$. Then, for each $x \sim \mathcal{D}_{\text{test}}$, an adapted parameter $\theta_x$ is obtained from $\theta$:

$$\theta_x \leftarrow \theta - \gamma \nabla_\theta \mathcal{L}_{\text{test}}(\theta, x)$$

The update above performs a single gradient-descent step; in practice it may use other optimizers and be applied over multiple steps. The prediction is then made as $\hat{y} \sim P_{\theta_x}(\cdot|x)$, and $\theta_x$ is discarded once the prediction is complete.

The objective $\mathcal{L}_{\text{test}}(\theta, x)$ takes different forms across the literature. We adopt perplexity minimization (Hu et al., 2025a;b), a representative TTT method, for its simplicity and effectiveness:

$$\theta_x \leftarrow \theta + \gamma \nabla_\theta \log(P_\theta(x))$$

## 2.2 Dataset

To better understand the behavior of TTT, we deliberately select datasets that resist the model's memorization. Specifically, we use the following three datasets.

**Memo-Trap** (McKenzie et al., 2024) tests a language model's instruction-following ability by asking it to complete a famous quote with an uncommon word. For example, the input `Write a quote that ends in the word "heavy": Absence makes the heart grow` expects the output "heavy" rather than "fonder". The dataset contains 860 examples.

**GSM-Symbolic** (Mirzadeh et al., 2024) perturbs GSM8k problems, for example by altering nouns and numbers, in order to test model robustness. It comprises two subsets: the easier "p1" (5,000 examples) and the harder "p2" (2,500 examples). We use the more challenging "p2" subset.

**Math-Perturb** (Huang et al., 2025) similarly perturbs MATH problems and consists of two subsets of 279 examples each, one simple and one hard. We experiment on the simple subset, as the hard subset is too difficult for the models. Because Llama performs poorly on this dataset (Huang et al., 2025), we evaluate only the Qwen 2.5 models on Math-Perturb.

## 2.3 Model

We use the Llama 3 (Dubey et al., 2024) and Qwen 2.5 (Qwen et al., 2025) model families. Specifically, we evaluate the 1B/3B/8B sizes of Llama 3 and the 1.5B/3B/7B sizes of Qwen 2.5. We use base models for Memo-Trap and instruction-tuned models for the other two datasets. For Llama, we use the 3.2 release for the 1B and 3B models and the 3.1 release for the 8B model.

## 2.4 Hyperparameters

Because hyperparameter robustness is central to this paper, we sweep the hyperparameters as broadly as possible to identify configurations that make the model robust. First, we sweep the learning rate and the number of training steps over a wide range: the learning rate spans five orders of magnitude (from 1$e$-8 to 1$e$-3), and the number of training steps reaches up to 100. Second, for the optimizer, regularization, momentum, and gradient normalization, we restrict ourselves to common values to avoid overfitting our conclusions to the chosen tasks. We compare the SGD and Adam optimizers; use weight decays of 0.01 and 0.5; set the momentum coefficient to 0.9; and use a gradient-normalization value of 10, since the more common value of 1 converges extremely slowly.

# 3 Main Observation 1: TTT Is Not A Panacea

**TTT works on AIME 2024 but struggles on AIME 2025.** As a first step, we replicate prior results on AIME to confirm that TTT is effective on popular benchmarks. Table 1 reports the performance of Qwen 2.5 on AIME with TTT and the SGD optimizer. Qwen 2.5 attains better and more stable performance with TTT than without it, consistent with prior observations (Hu et al., 2025b). On AIME 2025, however, the models do not improve under TTT. This contrast is informative because it functions as a natural, contamination-controlled comparison: Qwen 2.5 was released after AIME 2024 but before AIME 2025, so its training data can plausibly overlap with the former but not the latter, while the two test sets are matched in format and intended difficulty. The fact that TTT gains appear on the pre-release set and largely vanish on the post-release set is consistent with—though not by itself proof of—the contamination hypothesis. We treat it as the most controlled evidence we currently have, and discuss in Section 9.1 what a fully contamination-free evaluation would add.

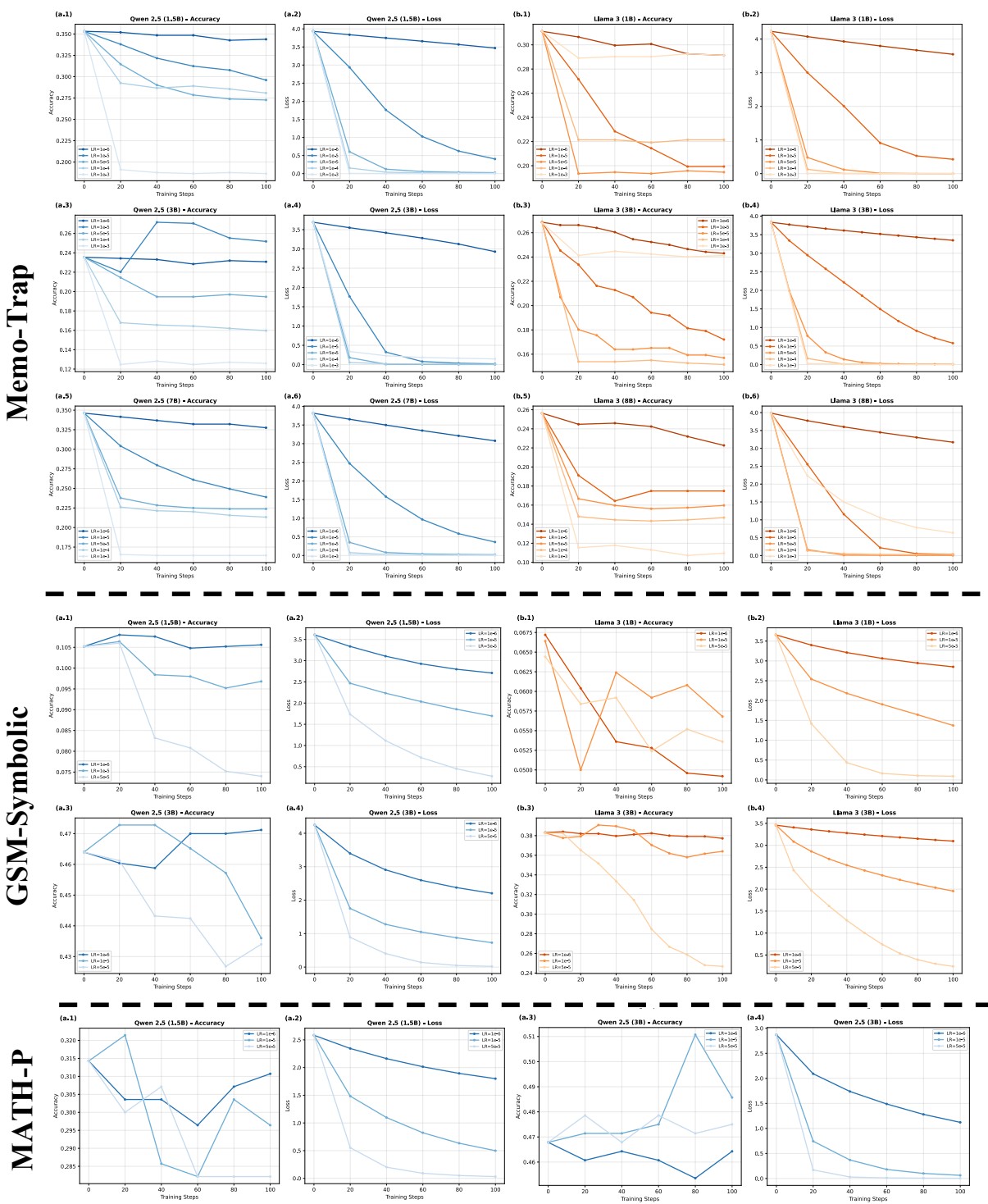

Figure 1: Qwen 2.5 and Llama 3 models across learning rates and numbers of training steps during TTT on the three datasets; the $x$-axis is the number of training steps. **Color:** blue denotes Qwen 2.5 and orange denotes Llama 3; darker shades indicate higher learning rates. **Rows:** model sizes. **Columns:** the first and third columns show accuracy, and the second and fourth show loss. Compared with the AIME results in Table 1, TTT on these datasets rarely achieves stable gains. Moreover, the learning dynamics vary across models, sizes, tasks, learning rates, and numbers of training steps, and no universal recipe emerges.

| Dataset | Model | Learning Rate | Training Steps | | | | | |
|---|---|---|---|---|---|---|---|---|
| | | | 0 | 20 | 40 | 60 | 80 | 100 |
| AIME 2024 | Qwen 2.5 1.5B-Instruct | $1e-6$ | 3.33 | 3.33 | 3.33 | **6.67** | **6.67** | **6.67** |
| | | $1e-5$ | 3.33 | **6.67** | **6.67** | **6.67** | **6.67** | 3.33 |
| | | $5e-5$ | 3.33 | 3.33 | 0.00 | 3.33 | 3.33 | **6.67** |
| | Qwen 2.5 3B-Instruct | $1e-6$ | 6.67 | **10.00** | 6.67 | 6.67 | 6.67 | 3.33 |
| | | $1e-5$ | 6.67 | **13.33** | 3.33 | 3.33 | **6.67** | 3.33 |
| | | $5e-5$ | **6.67** | 3.33 | 3.33 | 3.33 | 3.33 | 0.00 |
| AIME 2025 | Qwen 2.5 1.5B-Instruct | $1e-6$ | 3.33 | 3.33 | 3.33 | 3.33 | 3.33 | 3.33 |
| | | $1e-5$ | 3.33 | 3.33 | 0.00 | 0.00 | 0.00 | 0.00 |
| | | $5e-5$ | 3.33 | 0.00 | 0.00 | 0.00 | 0.00 | 0.00 |
| | Qwen 2.5 3B-Instruct | $1e-6$ | 6.67 | 6.67 | 3.33 | 3.33 | 3.33 | 3.33 |
| | | $1e-5$ | 6.67 | **10.00** | 3.33 | 3.33 | 0.00 | 0.00 |
| | | $5e-5$ | 6.67 | 3.33 | 6.67 | 6.67 | 6.67 | 3.33 |

Table 1: TTT with the SGD optimizer on AIME 2024 and AIME 2025. Qwen 2.5 improves on AIME 2024 but fails to achieve stable gains on AIME 2025.

**TTT struggles on anti-memorization datasets.** To examine this further, we run TTT with the SGD optimizer on the three anti-memorization datasets. Figure 1 shows accuracy and loss as a function of the number of training steps and learning rate. Overall, the models rarely improve under TTT. Qwen 2.5 1B and 7B show no gains on Memo-Trap, and even when the Qwen 2.5 models do improve, the gains are unstable and highly sensitive to the number of training steps. For example, Qwen 2.5 3B suffers a sharp performance drop on GSM-Symbolic once the number of training steps exceeds 40 at a learning rate of $1e$-5, and Qwen 2.5 1.5B drops sharply once the number of steps exceeds 20 at the same learning rate. Most Llama models also degrade under TTT. We note explicitly that these results establish *that* TTT fails to deliver reliable gains on anti-memorization tasks; they do not on their own establish *why*. An alternative to the contamination hypothesis is simply that these tasks are intrinsically harder for the base models, which could depress TTT gains independently of memorization. Disentangling these explanations requires either a contamination-free evaluation or diagnostics that separate recall from adaptation (Section 9.1); the AIME 2024/2025 contrast above is our closest current approximation.

**No universal recipe.** A general rule for choosing hyperparameters would considerably ease deployment, but Figure 1 shows that no such rule emerges. On GSM-Symbolic, a learning rate of $1e$-6 yields improvement for the Qwen 2.5 3B models when training 80 steps, whereas $1e$-5 quickly causes performance drops. On Math-Perturb, the opposite holds: $1e$-5 yields improvement for Qwen 2.5 3B for training 80 steps, whereas $1e$-6 leads to steady degradation. The optimal number of training steps likewise varies across datasets and models.

## 4 Observation 2: The Optimizer Has a Dramatic Effect

**SGD achieves better stability and performance than Adam.** Conventional wisdom in LM optimization favors the Adam optimizer. However, TTT typically operates on a single example, and we find that Adam often induces a problematic optimization landscape. Figure 2 shows TTT performance with Adam, where the green curve denotes TTT with SGD. Although Adam exhibits steep loss decay and converges within a few steps, its downstream performance is far worse than SGD's at the same or higher loss in most cases. SGD, by contrast, shows smooth loss decay and, more importantly, substantially more stable and higher performance. For example, on Memo-Trap and Math-Perturb, Qwen 2.5 3B exhibits nearly identical loss curves under Adam ($1e$-7) and SGD ($1e$-5) yet very different downstream performance. We attribute Adam's catastrophic drops to its adaptive estimation: as Algorithm 1 shows, the first Adam step degenerates into a sign method.

The sign method has two intuitive negative effects in TTT:

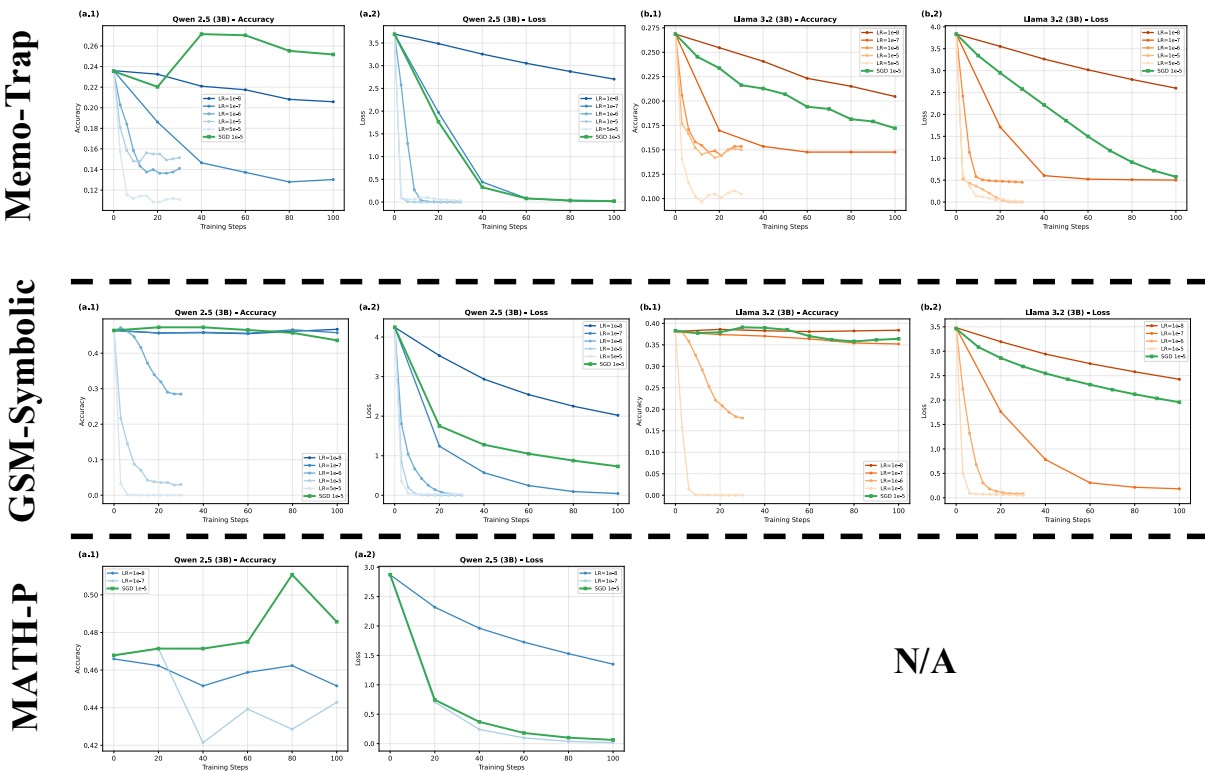

Figure 2: TTT results with the Adam optimizer; the figures follow the same structure and interpretation as Figure 1, and the green curve denotes TTT with SGD. Although Adam converges quickly, its final performance can be disastrous. SGD converges more slowly but performs much better at a comparable loss in most cases, indicating that Adam induces a problematic TTT loss landscape.

---

**Algorithm 1:** First Step of Adam Optimizer - Sign Method Behavior

---

**Require:** Learning rate $\alpha > 0$, exponential decay rates $\beta_1, \beta_2 \in [0, 1)$, small constant $\epsilon > 0$
**Require:** Initial parameter $\theta_0$, gradient $g_1 = \nabla f(\theta_0)$

1: **Initialize:** $m_0 = 0$, $v_0 = 0$ $\qquad\qquad\qquad\qquad\qquad$ ▷ First and second moment estimates
2: **Update biased first moment:**
3: $m_1 = \beta_1 \cdot m_0 + (1 - \beta_1) \cdot g_1 = (1 - \beta_1)g_1$
4: **Update biased second moment:**
5: $v_1 = \beta_2 \cdot v_0 + (1 - \beta_2) \cdot g_1^2 = (1 - \beta_2)g_1^2$
6: **Compute bias-corrected estimates:**
7: $\hat{m}_1 = \frac{m_1}{1 - \beta_1^1} = \frac{(1 - \beta_1)g_1}{1 - \beta_1} = g_1$
8: $\hat{v}_1 = \frac{v_1}{1 - \beta_2^1} = \frac{(1 - \beta_2)g_1^2}{1 - \beta_2} = g_1^2$
9: **Parameter update:**
10: $\theta_1 = \theta_0 - \alpha \cdot \frac{\hat{m}_1}{\sqrt{\hat{v}_1} + \epsilon} = \theta_0 - \alpha \cdot \frac{g_1}{\sqrt{g_1^2} + \epsilon}$
11: **When** $|g_1| \gg \epsilon$**:** $\theta_1 \approx \theta_0 - \alpha \cdot \text{sign}(g_1)$

---

- Distorted gradient direction: every coordinate of the gradient is rescaled to either $+1$ or $-1$.

- Excessively large updates: $+1$ and $-1$ are large step sizes in this regime.

Because Adam's performance remains low even at a learning rate of $1e\text{-}8$ (Figure 2), we believe the gradient direction is a key factor behind its collapse. We present this as a hypothesis rather than a verified mechanism. A clean test would also compare against Adam variants (e.g., a few initial SGD steps). These diagnostics would be the most direct way to confirm or falsify the explanation; we did not run them in this work and leave the concrete analysis as future work. We note that SGD reduces to gradient descent (GD) in the TTT setting; for consistency with common usage, we continue to refer to it as SGD.

A related strategy is to reduce the number of tunable parameters to slow Adam's fast convergence and potentially improve its performance. Unfortunately, Hu et al. (2025b) show that TTT performance with Adam still deteriorates dramatically after only a few additional training steps, even when the tunable parameter set is small.

**Loss is a weak indicator of performance.** Even when Adam attains loss curves similar to SGD's (e.g., Memo-Trap with Qwen 2.5 3B in Figure 2), the corresponding accuracy curves differ dramatically. Figure 1 shows the same pattern: Qwen 2.5 3B at learning rates $1e\text{-}4$ and $5e\text{-}5$ produces nearly identical loss curves but different accuracy, and Llama shows the same effect between $1e\text{-}3$ and $1e\text{-}4$. Loss alone is therefore an unreliable signal for judging whether the model is well adapted, even though it is often one of the few metrics available at deployment time.

Prior work (Sun et al., 2020) reports that, on certain vision tasks, Adam overfits quickly while SGD continues to improve performance. We show that this conclusion still holds in general for LMs.

## 5 Observation 3: Gradient Normalization Makes SGD More Robust

Although SGD performs well, it still overfits noticeably as the number of training steps grows. Simply lowering the learning rate to $1e\text{-}6$ is not a reliable remedy: performance may stagnate due to slow convergence, or even decline (e.g., Qwen 2.5 3B with SGD on Math-Perturb), as shown in Figure 1.

Three common strategies for controlling overfitting are regularization (e.g., weight decay), gradient normalization, and momentum. To avoid overfitting our hyperparameters to the test datasets, we consider only commonly used values.

**Regularization does not prevent overfitting.** Figure 3 shows the effect of weight decay on TTT. We test a value of 0.01, the default in PyTorch's AdamW implementation, and a larger value of 0.5. In both cases, weight decay has little effect: the learning curves nearly coincide with those of the no-weight-decay

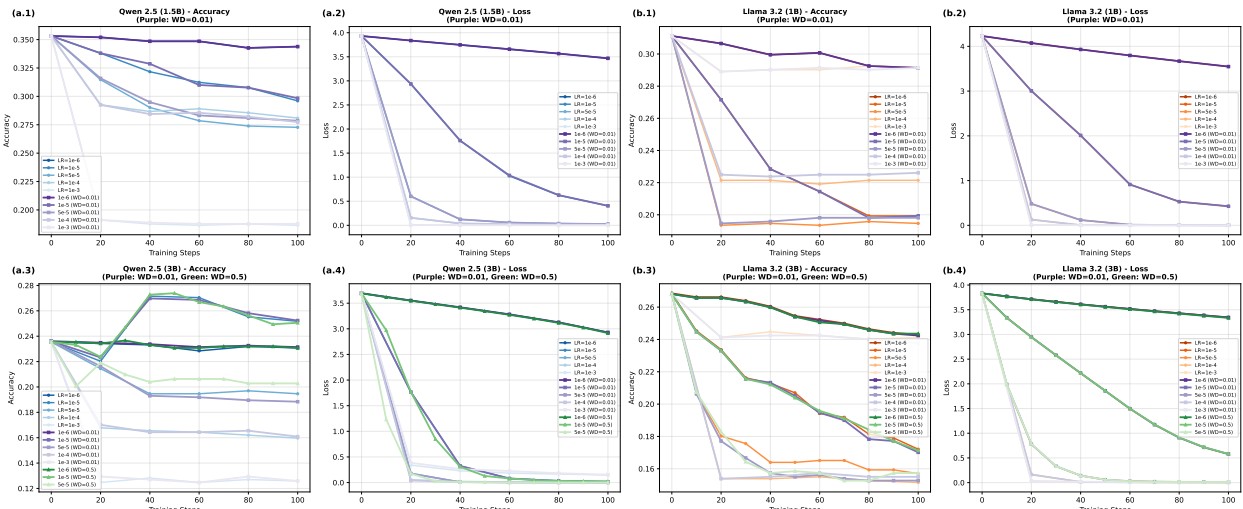

Figure 3: Effect of weight decay on TTT on the Memo-Trap dataset; the structure and interpretation match Figure 1. Within each model family, the two sizes share a color palette (blue or orange) to ease comparison across scales, and darker shades indicate higher learning rates. Purple curves with square markers denote weight decay 0.01 and green curves denote weight decay 0.5. Notably, the purple and green curves almost coincide with the blue curves (i.e., no weight decay).

baseline. This may indicate that TTT modifies only a small fraction of the parameters, which limits the influence of regularization.

**Gradient normalization slows overfitting.** Figure 4 shows the effect of gradient normalization. We use a normalization value of 10 rather than 1, as the latter converges too slowly. Unlike weight decay, gradient normalization has a clear effect: it generally mitigates catastrophic performance drops and sometimes improves peak performance. For example, on Memo-Trap, Qwen 2.5 3B reaches a worst-case accuracy of roughly 13% without gradient normalization but roughly 20% with it; moreover, two different learning rates with gradient normalization attain the same high accuracy, whereas without it only a single learning rate yields a gain after TTT. On GSM-Symbolic, Qwen 2.5 3B improves further when gradient normalization is combined with TTT. Gradient normalization is not universally beneficial, however, as its peak performance is occasionally lower than that of plain TTT. Since this paper is primarily concerned with the stability of TTT, gradient normalization is nonetheless an attractive choice.

**Momentum plays a role similar to gradient normalization but is less effective.** Figure 5 shows the effect of momentum on TTT. Its effect broadly resembles that of gradient normalization. However, because momentum occasionally produces an even more severe performance drop (e.g., Qwen 2.5 1.5B), gradient normalization is the preferable choice.

## 6 Observation 4: Tuning FFNs Yields Better Peak Performance, While Tuning Attention Yields Better Worst-Case Performance

A natural idea is to reduce the number of tunable parameters in order to mitigate overfitting. Prior work (Hu et al., 2025b) shows that overfitting persists even when the tunable parameter is only a vector in $\mathbb{R}^{1 \times d}$ applied on top of the final representation, where $d$ is the hidden dimension. Here, we instead ask which parameters contribute most to TTT. Specifically, we tune only the FFNs or only the attention modules and report the results in Figure 6. For Qwen 2.5 3B, the attention modules contain roughly 340M parameters and the FFNs roughly 2.4B; both are far more than sufficient to fit a single test-time example. For Llama 3, the curves match the conventional expectation that tuning FFNs overfits more readily. For Qwen 2.5, however, tuning the FFNs yields higher peak performance despite their larger size, suggesting a functional

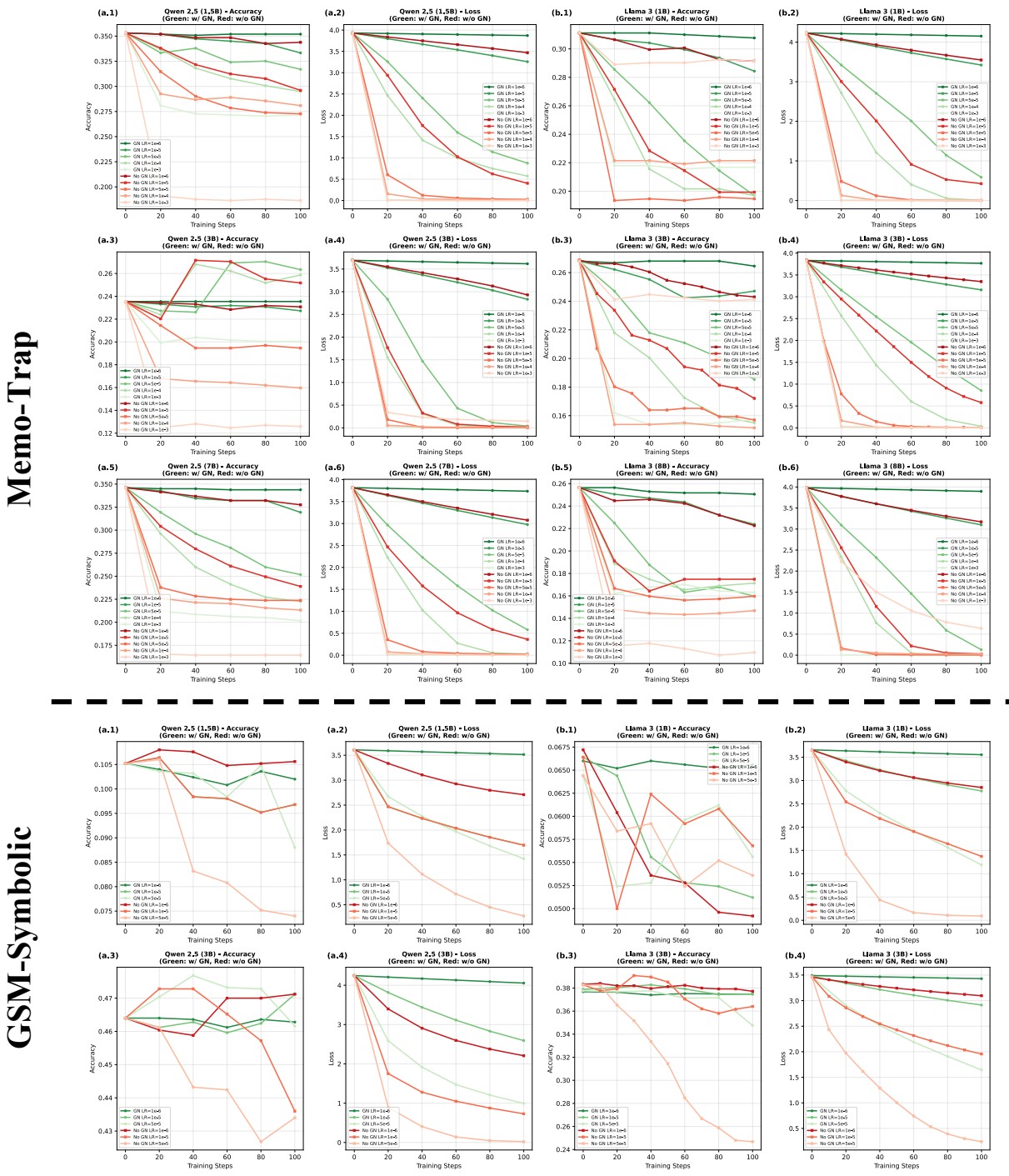

Figure 4: TTT with gradient normalization. Red and green curves denote TTT without and with gradient normalization, respectively; the structure and interpretation match Figure 1. The figure conveys two messages: (1) gradient normalization generally mitigates catastrophic performance drops, and (2) it sometimes yields better performance than the baseline. Although gradient normalization occasionally lowers peak performance, its stability makes it a good choice for TTT.

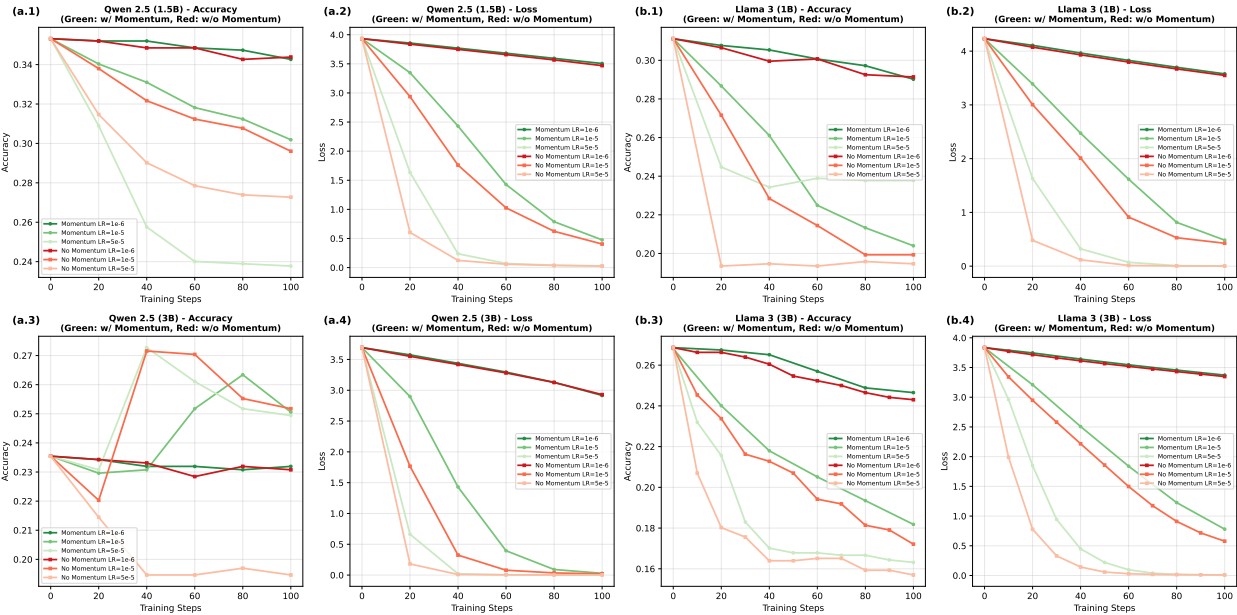

Figure 5: TTT with momentum on Memo-Trap. Red and green curves denote TTT without and with momentum, respectively; the structure and interpretation match Figure 1. Momentum plays a role similar to gradient normalization, mitigating catastrophic performance drops and producing a more stable TTT curve. Unlike gradient normalization, however, it sometimes leads to an even more severe drop (e.g., Qwen 2.5 1.5B).

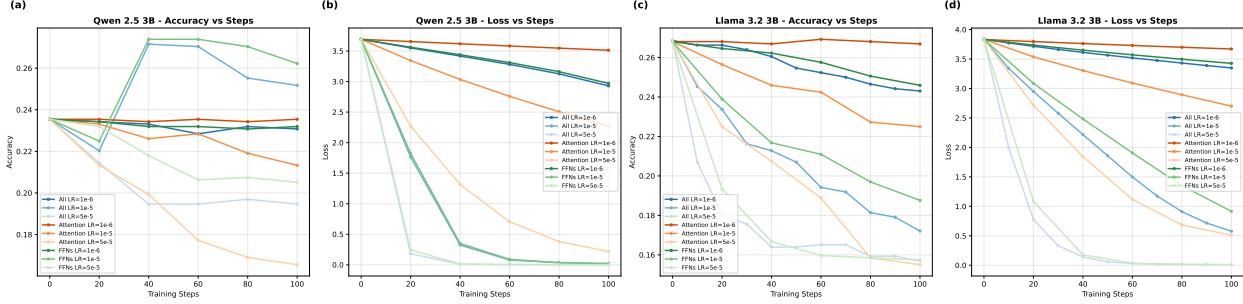

Figure 6: TTT performance when tuning FFNs or attention modules on the Memo-Trap dataset. Red, green, and blue curves denote tuning attention, FFNs, and all parameters, respectively. Although FFNs have many more parameters than attention modules, both are far more than sufficient to fit a single test-time example. Nevertheless, tuning FFNs achieves much higher peak performance than tuning attention modules, suggesting a functional difference between the two.

difference between FFNs and attention modules. One hypothesis is that FFNs act as a memory bank (Geva et al., 2021), which may make them better suited to TTT.

# 7 Computation and Latency Cost

Because TTT performs gradient updates at inference time, its cost differs fundamentally from standard decoding. For every test example, our setup performs up to 100 forward–backward update steps before a single prediction is produced, so the per-query cost is closer to that of a short fine-tuning run than to a forward pass. Three properties make this expensive in deployment: (i) the cost scales roughly linearly with the number of update steps, which our sensitivity analysis shows cannot be safely fixed to a small value across tasks; (ii) backward passes require activations and optimizer state to be held in memory, raising the memory footprint relative to inference-only serving; and (iii) the adapted parameters $\theta_x$ are discarded after each query, so the cost is paid per example and cannot be amortized across a batch of distinct inputs. We flag this qualitatively to make the practical trade-off explicit; we did not conduct a controlled latency and memory profiling study, and we regard a careful cost–accuracy characterization as an important piece of future work before TTT can be recommended for latency-sensitive deployment.

# 8 Related Work

**Test-time training.** Originally introduced for domain generalization (Sun et al., 2020), TTT updates model parameters using a self-supervised loss on each test example. Subsequent work extended this idea to masked autoencoders (Gandelsman et al., 2022), video streams (Wang et al., 2025), and multimodal generation (Dalal et al., 2025). Recent studies have adapted TTT to large language models, typically via input-perplexity minimization (Hu et al., 2025b;a) or RL-based reward optimization (Zuo et al., 2025), reporting gains on benchmarks such as GSM8K (Cobbe et al., 2021), MATH (Hendrycks et al., 2021a), and AIME. Akyürek et al. (2024) find that input-perplexity minimization is surprisingly effective for in-context learning: training on the few-shot examples before performing in-context learning yields a clear improvement over in-context learning alone. Despite these successes, popular benchmarks are known to suffer from possible data contamination (Wu et al., 2025), leaving the true generalization ability of TTT unclear. More recently, Sun et al. (2024) propose TTT-style layers whose hidden states are dynamically adapted by self-supervised learning on test sequences, showing that such layers reduce perplexity as more context is observed and pointing toward sequence-adaptive architectures. Hardt & Sun (2024) adopt a retrieval-based scheme in which, for each test example, the model retrieves a small set of nearest neighbors and performs a single gradient update on them before inference; this provides a practical TTT baseline for LMs while highlighting the importance of index quality and the risk of overfitting under adaptation.

**Optimization and stability.** Existing TTT approaches for LLMs commonly use the Adam optimizer (Kingma & Ba, 2014) with fixed hyperparameters inherited from fine-tuning practice (Hu et al., 2025b). Yet TTT operates in a distinct single-example optimization regime in which such settings may be suboptimal. Earlier work on vision tasks suggested that SGD can offer smoother convergence, but a systematic comparison for language models has been lacking. Moreover, although prior studies have explored regularization techniques such as weight decay and momentum to stabilize adaptation, their effectiveness for large-scale TTT remains limited (Sun et al., 2024).

**Parameter-efficient adaptation.** Prior work typically tunes all parameters of a language model during TTT. A complementary line of research on parameter-efficient fine-tuning (PEFT) adapts models by updating only a small subset of parameters, through mechanisms such as adapters (Houlsby et al., 2019), LoRA (Hu et al., 2022), and prompt or prefix tuning (Ding et al., 2023; Li & Liang, 2021). By decoupling adaptation capacity from full-model updates, these methods improve stability and efficiency, inspiring recent extensions of test-time adaptation that operate through adapter or prompt layers (Gao et al., 2022; Wang et al., 2022).

# 9   Conclusion and Discussion

This paper presents a preliminary but critical analysis of the robustness and practical applicability of input-perplexity-based test-time training for language models. Through extensive experiments on anti-memorization datasets across multiple model families and sizes, we report several findings that complicate the current understanding of TTT and suggest directions for future work.

First, TTT's effectiveness is not universal. While it improves performance on popular benchmarks such as AIME 2024, it struggles on tasks specifically designed to resist memorization, such as Memo-Trap, GSM-Symbolic, and Math-Perturb. This discrepancy raises the question of whether TTT primarily performs genuine adaptation or instead recalls memorized patterns; as discussed above, our experiments establish the failure but do not settle its cause.

Second, we identify a striking and previously underappreciated difference between optimizers in the TTT setting. Despite Adam's dominance in conventional LM training, SGD consistently outperforms it by a substantial margin under TTT. We hypothesize that this stems from Adam's sign-method-like behavior in single-example optimization, which distorts the gradient direction and destabilizes performance. This observation has immediate practical implications for TTT systems.

Third, we observe substantial hyperparameter sensitivity across models, model sizes, and tasks. No universal recipe exists for selecting the learning rate and number of training steps: a configuration that is optimal for one task may cause catastrophic drops on another. This poses a fundamental challenge for real-world deployment, where validation sets are typically unavailable for tuning at test time.

Fourth, among the regularization strategies we consider, gradient normalization is the most effective for improving TTT robustness. It mitigates catastrophic performance drops, reduces sensitivity to the learning rate, and sometimes improves peak performance. Weight decay, despite its prevalence in standard training, has little effect on overfitting in TTT, and momentum offers benefits similar to gradient normalization but is less reliable.

Finally, our analysis of which components to tune reveals consistent patterns. Tuning only the feed-forward networks can match or exceed full-model fine-tuning, whereas tuning only the attention modules, yields more stable worst-case performance. These observations suggest functional differences between the two components that merit further study.

## 9.1   Limitations and Future Work.

This study is a preliminary analysis and has several limitations. First, our experiments cover three specific anti-memorization tasks, which, while informative, may not represent the full range of real-world applications. Second, we restrict the hyperparameters to common values to avoid overfitting to our test scenarios; a more exhaustive search might reveal additional effects. Third, our account of why SGD outperforms Adam is incomplete: we identify gradient direction as a likely factor, but a deeper theoretical understanding is still needed. Fourth, and most importantly for the scope of our conclusions, we study a single TTT objective—input perplexity minimization—so our findings may not transfer to other TTT formulations. We make this scope limitation explicit: because input perplexity minimization is only one of several TTT objectives (e.g., RL-based reward optimization or reconstruction-based losses), the failure modes and remedies we report should be read as evidence about this objective specifically rather than about test-time training in general. A broader study spanning multiple objectives is an important direction that we do not undertake here. Fifth, we consider only single-example optimization and do not incorporate additional examples, a setting that is closer to domain adaptation.

Three further limitations temper how our claims should be read. **(1) Decoding method.** The results we report are from single runs (greedy decoding) instead of multiple runs (temperature-based decoding). **(2) Failure versus contamination.** Our experiments establish that TTT fails to give reliable gains on anti-memorization tasks, but they do not isolate contamination as the cause; we therefore treat contamination as a hypothesis rather than a finding. We did not run a dedicated contamination-free evaluation (e.g., problems generated after the model release date, or synthetic low-overlap problems) or a recall-versus-adaptation

diagnostic; the AIME 2024/2025 contrast in Section 3 is our closest available approximation, and a controlled study is left to future work. **(3) Optimizer mechanism.** As discussed in Section 4, the sign-method explanation for Adam's collapse is a hypothesis we did not verify; confirming it would require measurable diagnostics (e.g., Adam-variant comparisons), which we leave to future work.

Finally, we delineate which of our conclusions are likely specific to the studied algorithm versus broadly applicable to test-time adaptation. Conclusions that are tied to input-perplexity minimization include the specific failure pattern on anti-memorization tasks and the claim that loss is a weak proxy for accuracy under this objective. Conclusions we expect to transfer more broadly—because they stem from the single-example, no-validation-set optimization regime common to most TTT methods—include the strong hyperparameter sensitivity, the absence of a universal recipe, and the stabilizing role of gradient normalization. We frame the broader claims as hypotheses for objectives such as RL-based or reconstruction-based TTT, which we did not test.

As test-time training becomes a new paradigm for scaling LMs, we believe specialized optimizers for TTT are one promising direction.

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
