# OpenReview forum: "When Test-Time Training Fails: A Critical Analysis of Robustness and Hyperparameter Sensitivity"
_TMLR — Accepted by TMLR_

### Review · Reviewer_jAMV · 2025-12-29

**Summary Of Contributions:**

This paper provides a careful, negative-result-heavy analysis of test-time training (TTT) via input perplexity minimization. Its core contribution is the evidence that current TTT is brittle, hyperparameter-sensitive, and unreliable on tasks that resist memorization. This is a valuable reality check on a technique that has been gaining hype based on benchmark wins.

Strengths:
1. Focus on anti-memorization tasks (major strength)
Evaluating on Memo-Trap, GSM-Symbolic, and Math-Perturb is exactly what’s missing from much prior TTT work. This design directly challenges the claim that TTT performs “online adaptation” rather than exploiting memorized patterns. The failure of TTT on these datasets strongly suggests benchmark gains may be artifact-driven, not capability-driven. In the real world, inputs are often distribution-shifted and adversarial, not benchmark-like.

2. Hyperparameter sensitivity analysis is thorough
Wide sweeps over learning rates (5 orders of magnitude), training steps, optimizers, and normalization. The paper explicitly demonstrates no universal recipe across models or tasks. It also shows that loss is a misleading signal, which is critical since loss is often the only observable metric at inference time.

3. Optimizer findings are actionable
Clear evidence that SGD outperforms Adam in TTT, despite Adam’s dominance in LM training. The argument that Adam’s adaptive steps distort gradient direction in single-sample regimes is plausible and concerning.

4. Gradient normalization as a practical stabilizer
Gradient norm clipping materially improves robustness and reduces catastrophic drops. Even when peak performance doesn’t improve, worst-case behavior improves. Identifies a concrete knob that meaningfully improves model deployability.

Limitations:
1. Narrow scope: Only one TTT objective
The study focuses exclusively on input perplexity minimization.

2. Computational and latency costs are underexplored
Multiple gradient steps per query (up to 100) is expensive.

**Additional Comments:**

Sharing reproducible code will make this paper more impactful

**Audience:**

Yes

**Audience Explanation:**

The experiments are well designed and have actionable insights. We need evaluations that mimic real world scenario and this paper is a good one in that direction

**Broader Impact Concerns:**

-

**Claims And Evidence:**

Yes

**Claims Explanation:**

The authors show the results on 3 good datasets and perform good hyperparameter sweeps. The experiments were conducted on a good range of models. The results and algorithm is well documented

**Requested Changes:**

A note on computation & latency costs would make the paper stronger

---

### Review · Reviewer_Msof · 2026-01-01

**Summary Of Contributions:**

This paper studies test-time training (TTT) for LLMs via input perplexity minimization at inference time, with the goal of characterizing generalization and hyperparameter sensitivity when no validation signal is available. The authors evaluate Qwen 2.5 and Llama 3 on Memo-Trap, GSM-Symbolic p2, and Math-Perturb, and run broad sweeps over learning rate and number of update steps. They further ablate optimizer choice (SGD vs Adam), weight decay/momentum, gradient normalization, and which parameter subsets are updated (FFN vs attention).

Key empirical findings include:
(i) Reported gains from perplexity-based TTT do not transfer reliably across datasets and can change substantially across closely related test sets.
(ii) In the tested regime, SGD is generally more stable than Adam, even when Adam reduces perplexity quickly.
(iii) Weight decay has limited impact on downstream accuracy in the reported experiments.
(iv) Gradient normalization can reduce catastrophic performance drops and reduce sensitivity to tuning, but is not uniformly beneficial.
(v) Updating FFN parameters tends to improve best-case performance, while attention-only updates can improve worst-case stability.

**Audience:**

Yes

**Audience Explanation:**

TTT/TTL for LLMs is an active direction, with prior work proposing input-perplexity minimization as a test-time learning objective. This paper is valuable as a stress test of robustness and tuning sensitivity under settings designed to resist memorization.

**Broader Impact Concerns:**

no concern

**Claims And Evidence:**

No

**Claims Explanation:**

The simulations support the narrow claims about sensitivity and optimizer/normalization effects. But broader interpretations (e.g., that observed gains are due to memorization/contamination rather than adaptation) are not directly tested. The proposed explanation for Adam’s failure is also not empirically validated (no diagnostics such as step norms, gradient alignment, parameter drift, or comparisons to Adam variants). Stability claims need multi-seed results. The stronger causal narrative, that TTT gains reflect memorization or data contamination rather than adaptation, remains under-tested, as the study does not include a controlled, contamination-free evaluation or diagnostics that distinguish recall from genuine adaptation.

**Requested Changes:**

1-Separate “TTT fails on anti-memorization tasks” from “TTT relies on contamination.” Add at least one contamination-minimized evaluation (e.g., post-cutoff synthetic generation or carefully constructed problems with low overlap risk) and/or diagnostics that distinguish rote recall from adaptation.

2-Report variance across runs. Provide multiple seeds (or at minimum variability summaries) for the main claims about “catastrophic drops” and “stability,” since the paper emphasizes sensitivity.

3-Strengthen the optimizer mechanism claim. The “better gradient direction” hypothesis for SGD needs measurable evidence (e.g., gradient alignment/step statistics) rather than a qualitative explanation.

---

### Review · Reviewer_E2RW · 2026-01-31

**Summary Of Contributions:**

This paper provides an analysis of test-time training for LLM.
This work contributes new and practically relevant insights into when TTT works and when it fails. By evaluating TTT on several anti-memorization datasets, the authors highlight that previously reported gains may not generalize beyond popular benchmarks. This perspective offers new knowledge for the community regarding the limits of TTA in LLMs.
The paper conducts extensive sweeps over learning rates, training steps, optimizers, and regularization techniques across multiple model families and sizes. Overall, the work delivers a good empirical investigation that is likely to be useful for researchers and practitioners considering TTT in real-world settings.

**Audience:**

Yes

**Audience Explanation:**

TTT is rapidly growing research directions in the LLM community, and many recent works report strong improvements on popular benchmarks. This paper addresses an important complementary question: under what conditions TTT does not work reliably, and how sensitive it is to optimization choices.

**Claims And Evidence:**

Yes

**Claims Explanation:**

The main claims of the paper are generally supported by empirical evidence. The authors evaluate TTT across multiple model families, model sizes, datasets, and many configurations. In particular, the comparison between standard benchmarks and anti-memorization datasets directly supports the claim that TTT gains do not always generalize. The extensive ablations on optimizers, learning rates, training steps, gradient normalization, and parameter subsets further strengthen the analysis of hyperparameter sensitivity and stability.

**Requested Changes:**

- Several figures contain small fonts, dense legends, and light-colored curves that are difficult to distinguish, especially when printed or viewed at standard zoom.
- Since this paper focus on a specific TTT formulation, a short discussion clarifying which conclusions are likely algorithm-specific versus more broadly applicable to test-time adaptation would help position the results within the wider TTT literature.

---

### Decision · Action_Editor_36CZ · 2026-04-20

**Recommendation:** Accept with minor revision

**Audience:**

Yes

**Audience Explanation:**

The paper will attract some audience in LLMs.

**Claims And Evidence:**

Yes

**Claims Explanation:**

This paper offers a strong empirical analysis of test-time training (TTT) for LLMs, highlighting its brittleness, hyperparameter sensitivity, and limited generalization to anti-memorization tasks. The experimental findings are valuable, especially regarding optimizer choice and stability. Reviewers believe that some causal claims (e.g., memorization vs. adaptation) are not fully supported, and variability across runs is not reported. Minor revisions should clarify the scope of conclusions, moderate unsupported interpretations, add variance estimates, and improve figure readability.